# Online Dynamics Learning for Predictive Control with an Application to Aerial Robots

**Tom Z. Jiahao**[*]**, Kong Yao Chee**[*]**, M. Ani Hsieh**
GRASP Laboratory
University of Pennsylvania United States
{zjh, ckongyao, m.hsieh}@seas.upenn.edu

**Abstract:** In this work, we consider the task of improving the accuracy of dynamic models for model predictive control (MPC) in an online setting. Although prediction models can be learned and applied to model-based controllers, these models are often learned offline. In this offline setting, training data is first collected and a prediction model is learned through an elaborated training procedure. However, since the model is learned offline, it does not adapt to disturbances or model errors observed during deployment. To improve the adaptiveness of the model and the controller, we propose an online dynamics learning framework that continually improves the accuracy of the dynamic model during deployment. We adopt knowledge-based neural ordinary differential equations (KNODE) as the dynamic models, and use techniques inspired by transfer learning to continually improve the model accuracy. We demonstrate the efficacy of our framework with a quadrotor, and verify the framework in both simulations and physical experiments. Results show that our approach can account for disturbances that are possibly time-varying, while maintaining good trajectory tracking performance.

**Keywords:** Online Learning, Model Learning, Model Predictive Control, Aerial Robotics

## 1 Introduction

In recent years, model predictive control (MPC) has shown significant potential in robotic applications. As an optimization-based approach that uses prediction models, the MPC framework leverages readily available physics models or accurate data-driven models to allow robotic systems to achieve good closed-loop performance for a variety of tasks. However, the reliance on accurate dynamic models makes it a challenging task for the controller to adapt to system changes or environmental uncertainties. In the case where the robot dynamics change during deployment or in the presence of disturbances, it becomes essential for the controller to update and adapt its dynamic model in order to maintain good performance. Recent advancements in deep learning techniques have demonstrated promising results in using neural networks to model dynamical systems. One advantage of neural networks over physics models is that they alleviate the need for bottom-up construction of dynamics, which often requires expert knowledge or physical intuition. Neural networks are also increasingly faster to optimize, owing to the development of optimization algorithms. This makes the application of neural networks to model predictive controllers more amenable, which allows the controllers to adapt to disturbances or system changes by refining its dynamic model. In this work, we propose a novel online dynamic learning algorithm using a deep learning method, knowledge-based neural ordinary differential equations. Through both simulations and physical experiments, we show that our framework, which comprises of a suite of algorithms, can improve the adaptiveness of MPC by continually refining its dynamic model of a quadrotor system, and in turn maintains desirable closed-loop performance during deployment.

**Related Work.** Traditional supervised machine learning paradigms consist of two phases – training and inference. The training phase requires data to be collected in advance and then takes place in

---

[*]Equal contribution, co-first authors.

6th Conference on Robot Learning (CoRL 2022), Auckland, New Zealand.

an offline fashion. After the model is trained, it is then deployed for inference. Online learning, on the other hand, performs learning with data arriving in a sequential order. Notably, online learning has been used to transfer existing knowledge to assist training [1, 2], predict general time series data [3, 4], and learn inverse dynamics in a derivative free manner [5].

Learning dynamic models has gathered increasing attentions because of the development in scientific machine learning. A spectrum of techniques has emerged ranging from blackbox approaches [6, 7], to methods that require some known structures [8, 9]. In the middle of this spectrum, techniques have been developed to combine physics knowledge with machine learning [10, 11, 12, 13], and physical priors have also been incorporated into neural networks [14]. A library based on continuous-time deep learning techniques have been developed for scientific machine learning [15]. Online model learning is increasingly being used by model-based control methods for their adaptability to changing dynamics. It has been used to learn dynamics models for a variety of robots including quadruple robots, robotic arms, and autonomous racing vehicles [16, 17, 18, 19]. However, using neural networks to model dynamics faces several challenges in robotic applications. Optimizing neural networks online can be slow for real-time robotic applications [20]. Hence, many existing works use other parametric models such as matrices [16] and Gaussian mixture models [17]. Within our framework, we make use of lightweight neural networks to learn residual dynamics, which allow the neural networks to be optimized online. In addition, a limited amount of data may be available for training during robot deployment. To address this problem, neural networks have been bootstrapped using previously recorded data for initial training [17], and local approximators are used to run in parallel to generate data for training neural networks. In model-based reinforcement learning (MBRL), the control policy and system dynamics are often jointly optimized [21]. However, the online updates can be slow, and therefore real-time application to robotic platforms with fast dynamics may be challenging. An application of MBRL to train a quadcopter requires three minutes of flight time [22], which may not be available during deployment. For our framework, through simulations and physical experiments, we show that our proposed framework only requires a small amount of data to train each model and it achieves good trajectory tracking performance.

There are a number of works that use data-driven models within an MPC framework. In [23, 24], Gaussian processes are used to characterize dynamics models for MPC. In [25], the authors use a hybrid model that combines a first-principles model with a neural network. Our work is hugely inspired by [25], but instead of learning the model offline, we devise a framework that allow the dynamic model to be continually improved during deployment. The motivation for online dynamics learning is so that the model is able to account for uncertainty and disturbances that are not captured by the training data. To the authors' best knowledge, there has yet been online learning techniques specifically developed for learning dynamical systems and being applied to model predictive control schemes. There are works in the literature that promote adaptiveness for MPC. In [26], the authors propose updating the state and control weights in the MPC optimization problem in an online setting. The authors in [27] propose an adaptive cruise control scheme that includes a real-time weight tuning strategy, where the weights in the optimization problem are adjusted based on various operating conditions. In our work, instead of optimizing for the cost weights or other parts of the objective function during deployment, we use deep learning to compensate for the uncertain dynamics that are not accounted for in the dynamics constraints within the MPC framework. In [28], a nonlinear MPC scheme augmented with a L1 adaptive component is proposed. The errors between the state estimator and L1 observer are incorporated into the control scheme through an adaptation law. In a similar vein, the authors in [29] proposed a linear MPC scheme combined with a L1 adaptive module for quadrotor trajectory tracking. Joshi et al. [30] use a learning rule to update weights of the outermost layer of the neural network. This network is then used in the adaptive term in the proposed controller for a quadrotor. In [31], a L1 adaptive control augmentation is incorporated into a geometric controller. In contrast to the above works, instead of augmenting existing controllers with an adaptive module, our proposed framework directly updates the dynamic constraints within the model predictive controller and the control inputs are obtained by solving the optimization problem.

## 2  Problem Formulation

Given a robot that uses a model predictive controller, we aim to use data to continually improve its closed-loop performance in an online setting. Specifically, we seek to refine the dynamics model within the MPC framework during robot deployment. We consider a robot with the following dy-

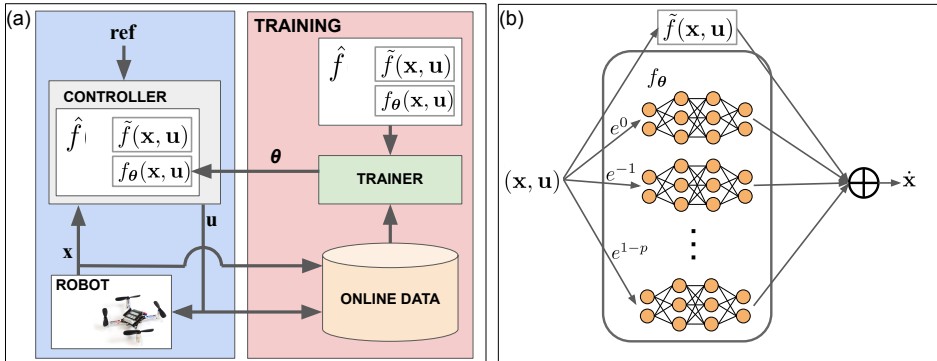

Figure 1: **Framework overview**. (a) The training process runs in parallel with the robot carrying out some tasks. The robot states and control inputs are collected for training. The trained model is used to update the dynamic model in the controller. (b) the neural network consists of a nominal model and repeatedly added neural networks. Each neural network and the nominal model run in parallel with each other, with their output coupled to give the state derivative.

namics,

$$\dot{\mathbf{x}} = f(\mathbf{x}, \mathbf{u}), \tag{1}$$

where the function $f$ represents the true dynamics of the robot and the vectors $\mathbf{x}$ and $\mathbf{u}$ are the state and the control input of the robot. As the robot is deployed for a specified task, a streaming sequence of data samples that consist of states and control inputs is collected. This is denoted by $\mathcal{S} := [(\mathbf{x}(t_0), \mathbf{u}(t_0)), (\mathbf{x}(t_1), \mathbf{u}(t_1)), \cdots]$, where $\{t_0, t_1, \cdots\}$ are the time stamps in which the data is collected. With $\mathcal{S}$, we seek to generate accurate estimates of the robot dynamics denoted by

$$\dot{\mathbf{x}} = \hat{f}(\mathbf{x}, \mathbf{u}), \tag{2}$$

where $\hat{f}$ represents an updated estimate of the dynamics model. This updated model $\hat{f}$ is then integrated back into the MPC framework to provide more accurate predictions of the dynamics, which in turn improves closed-loop performance. In particular, to construct $\hat{f}$ accurately, we utilize knowledge-based neural ordinary differential equations (KNODE) and tools from online learning.

## 3 Online Dynamics Learning

### 3.1 Knowledge-based Neural Ordinary Differential Equations

Neural ordinary differential equations (NODE) was first proposed to approximate continuous-depth residual networks [6]. It has since been used in scientific machine learning to model a wide variety of dynamical systems [11]. Knowledge-based neural ordinary differential equations (KNODE) is an extension of NODE that couples physics knowledge with neural networks by leveraging its compatibility with first principles knowledge, and has demonstrated learning continuous-time models with improved generalizability. Three aspects of KNODE make it a suitable candidate for our online learning task. First, it requires less data for training, which means each data collection period during robot deployment can be short, and thus improving the adaptiveness of the MPC. Second, KNODE is a continuous-time dynamic model, which is compatible with many existing MPC frameworks. Third, many robotic systems have readily available physics models that can be used as knowledge. The original KNODE was applied only to non-controlled dynamical systems, but variants of it have been developed to incorporate control inputs to dynamic models [25]. In this work, we use a similar approach as [25], where we concatenate the state and control as $\mathbf{z} = [\mathbf{x}^T, \mathbf{u}^T]^T$. During training, the control is simply ignored when computing loss.

Using KNODE model, the dynamics in (2) is expressed as $\hat{f}(\mathbf{z}, t) = M_\psi(\tilde{f}(\mathbf{z}, t), f_\theta(\mathbf{z}, t))$, where $\tilde{f}$ is the physics knowledge, $f_\theta$ is a neural network parametrized with $\theta$, and $M_\psi$ is a selection matrix parametrized with $\psi$, which couples the neural network with knowledge. A loss function is then

given by

$$\mathcal{L}(\theta, \psi) = \frac{1}{m-1} \sum_{i=1}^{m-1} \int_{t_i}^{t_{i+1}} \delta(t_s - \tau) \|\hat{\mathbf{x}}(\tau) - \mathbf{x}(\tau)\|^2 d\tau + \mathcal{R}(\theta, \psi), \tag{3}$$

where $m$ is the number of points in the training trajectory, $\delta$ is the Dirac delta function, $t_s \in T$ is any sampling time in $T$, and $\mathcal{R}$ is the regularization on the neural network and coupling matrix parameters. The estimated state $\hat{\mathbf{x}}(\tau)$ in (3) comes from $\hat{\mathbf{z}}(\tau)$, which is given by

$$\hat{\mathbf{z}}(\tau) = \mathbf{z}(t_i) + \int_{t_i}^{\tau} \hat{f}\left(\mathbf{z}(\omega), \omega\right) d\omega. \tag{4}$$

Intuitively, $\hat{\mathbf{z}}(\tau)$ is the state at $t = \tau$ generated using $\hat{f}$ with the initial condition $\mathbf{z}(t_i)$, and the loss function (3) computes the mean squared error between the one-step-ahead estimates of the states and the ground truth data. The integration in (4) is computed using numerical solvers in practice.

The optimization of the neural network parameters in KNODE can be done with either backpropagation or the adjoint sensitivity method, which is a memory efficient alternative to backpropagation. In this work, the adjoint sensitivity method is used, similar to [6]. In the following sections, the dynamic models are all based on KNODE.

## 3.2 Online Data Collection and Learning

| **Algorithm 1** Data collection and model updates | **Algorithm 2** Online dynamics learning |
|---|---|
| 1: Initialize the current time, last save time, total duration, and the collection interval as $t_i$, $t_s$, $t_N$, and $t_{col}$ | 1: Initialize the current time and total duration as $t_i$ and $t_N$ |
| 2: $t_i \leftarrow 0$ | 2: $t_i \leftarrow 0$ |
| 3: OnlineData $\leftarrow$ [] | 3: **while** $t_i < t_N$ **do** |
| 4: **while** $t_i < t_N$ **do** | 4:     **while** No new data available **do** |
| 5:     **if** New model is available **then** | 5:         Wait |
| 6:         Controller updates new model | 6:     **end while** |
| 7:         $t_s \leftarrow t_i$ | 7:     Train a new model with the newest data |
| 8:     **end if** | 8:     Save the trained model |
| 9:     **if** $t_i$ is not 0 and $t_i - t_s == t_{col}$ **then** | 9:     $t_i \leftarrow$ current time |
| 10:         Save OnlineData | 10: **end while** |
| 11:         $t_s \leftarrow t_i$ | |
| 12:         OnlineData $\leftarrow$ [] | |
| 13:     **end if** | |
| 14:     Robot updates state using control input | |
| 15:     Append new robot state and control input to OnlineData | |
| 16:     $t_i \leftarrow$ current time | |
| 17: **end while** | |

The basic features of the proposed online dynamics learning algorithm include the dynamic model update logic and a trainer that runs in parallel with the robot, which are described in Alg. 1 and 2 respectively.

For data collection, the robot state is repeatedly appended to a data array during deployment. This data array is saved and reset (Line 9 and 10 of Alg. 1 ) at regular time intervals (Line 8 of Alg. 1). A key hyperparameter of data collection is the collection interval $t_{col}$, which dictates how much data in a batch (measured in seconds) to be collected for training. With a longer collection interval, the adaptiveness of the algorithm will decrease because it not only takes longer to collect data but also takes more time to train a new model. This leads to less frequent model updates for the controller, and therefore less adaptiveness to system/environment changes. With a shorter collection interval, while the adaptiveness improves, a model is more likely to overfit during training due to the smaller training data size.

When new data becomes available, a model is trained as described in Alg. 2. A challenge for online dynamic model learning is how to continually improve the model with newly collected data. When the model gets updated, the closed-loop trajectory of the controller changes accordingly and any new data collected will reflect the updated controller. As a result, the previous dynamic model needs to be preserved, and training the neural network already onboard will not serve the purpose as it alters the previous controller. To tackle this problem, we use a fixed-sized queue, and repeatedly add new neural networks to the queue. We weigh the neural networks with exponential forgetting factors, and discard the oldest neural network to maintain the queue size as illustrated in Fig. 1 (b). This

design achieves constant computation and memory cost. Mathematically, given a nominal model $\tilde{f}$ as the first dynamic model estimate, the approximate model $\hat{f}$ is recursively constructed by

$$\hat{f}^{(i+1)} = M_{\psi_{(i+1)}}(\hat{f}^{(i)}, e^{i+1-p} f_{\theta_{(i+1)}}) \text{ for } i < p,$$
$$\hat{f}^{(0)} = \tilde{f}, \tag{5}$$

where $p$ is the queue size, the index $(i+1)$ denotes the $(i+1)$th update to dynamic model, and $f_{\theta_{(i+1)}}$ is the $i$th neural network added to the queue, parametrized by $\theta_{(i+1)}$. In this work, $M_\psi$ is formed by stacking two $n$ by $n$ identity matrices, where $n$ is the dimension of the state. For the $(i+1)$th update, only $\theta_{(i+1)}$ are trained, while other existing neural networks are frozen.

Also note that after each controller update, the next set of data for training must be entirely produced using the updated controller. If a set of data is collected using a mix of old and new controllers, it must be discarded so that the next set of training data represented the most up-to-date controller. This requirement is illustrated by the update in Alg. 1 Line 7 and the loop guard in Alg. 2 Line 4.

### 3.3 Applying Learned Models in MPC

Inspired by the framework in [25], we apply the updated dynamic model $\hat{f}$ in an MPC framework. Specifically, we solve the following constrained optimization problem in a receding horizon manner,

$$\underset{\substack{x_0,\ldots,x_N, \\ u_0,\ldots,u_{N-1}}}{\text{minimize}} \quad \sum_{i=1}^{N-1} \left( x_i^T Q x_i + u_i^T R u_i \right) + x_N^T P x_N \tag{6a}$$

$$\text{subject to} \quad x_{i+1} = f(x_i, u_i), \quad \forall i = 0, \ldots, N-1, \tag{6b}$$
$$x_i \in \mathcal{X}, \quad u_i \in \mathcal{U}, \quad \forall i = 0, \ldots, N-1, \tag{6c}$$
$$x_0 = x(t), \quad x_N \in \mathcal{X}_f, \tag{6d}$$

where $x_i, u_i$ are the predicted states and control inputs, $N$ is the horizon and $\mathcal{X}, \mathcal{U}, \mathcal{X}_f$ are the state, control input and terminal state constraint sets. $f(\cdot, \cdot)$ in (6b) is a discretized version of the learned KNODE model, as described in Sections 3.1 and 3.2. This model is used to predict the future states within the horizon. A precise model gives more accurate predictions of the states, which in turn provides more effective control actions. The weighting matrices $Q$ and $R$ are designed to penalize the states and control inputs within the cost function and $P$ is the terminal state cost matrix. $x(t)$ is the state obtained at time step $t$, which acts as an input to the optimization problem (6). Upon solving (6), the first element in the optimal control sequence $u_0^*$ is applied to the robot as the control action. The robot moves according to this control action and generate new state measurements $x(t + 1)$, which will be used to solve (6) at the next time step. The optimization problem is implemented and solved using CasADi [32]. An interior-point method IPOPT [33] within the library is used to solve the problem. The solver is warm-started at each time step by providing it with an initial guess of the solution, which is based on the optimal solution, both primal and dual variables, obtained from the previous time step.

## 4 Simulations

To demonstrate the efficacy of the framework in real-world robotic applications, we apply it to a quadrotor system and conduct tests in both simulations and physical experiments.

### 4.1 Dynamics of a Quadrotor System

To apply the KNODE-MPC-Online framework, we first construct a KNODE model, by combining a nominal model derived from physics, with a neural network. For the quadrotor, the nominal model can be derived from its equations of motion,

$$m\ddot{\mathbf{r}} = m\mathbf{g} + \mathbf{R}\eta, \quad \mathbf{I}\dot{\boldsymbol{\omega}} = \boldsymbol{\tau} - \boldsymbol{\omega} \times \mathbf{I}\boldsymbol{\omega}, \tag{7}$$

where $\mathbf{r}$ and $\boldsymbol{\omega}$ are the position and angular rates of the quadrotor, $\eta, \boldsymbol{\tau}$ are the thrust and moments generated by the motors of the quadrotor. $\mathbf{g}$ is the gravity vector and $\mathbf{R}$ is the transformation matrix

that maps $\eta$ to the accelerations. $m$ and $\mathbf{I}$ are the mass and inertia matrix of the quadrotor. Furthermore, by defining the state as $\mathbf{x} := [\mathbf{r}^\top \ \dot{\mathbf{r}}^\top \ \mathbf{q}^\top \ \boldsymbol{\omega}^\top]^\top$ and control input as $\mathbf{u} := [\eta \ \boldsymbol{\tau}^\top]^\top$, where $\mathbf{q}$ denotes the quaternions representing its orientation, the nominal component of the KNODE model can then be expressed as $\tilde{f}(\mathbf{x}, \mathbf{u})$. The KNODE model is then constructed using the algorithms described in Section 3.2.

## 4.2  Simulation Setup

A simulation environment for the quadrotor system is implemented based on the equations of motion given in (7). An explicit Runge-Kutta method (RK45) with a sampling time of 2 milliseconds is used for numerical integration to simulate dynamic responses of the quadrotor. The predictive controller described in Section 3.3 is assumed to have full measurements of the dynamics of the quadrotor. The solution of (6) provides control commands to the quadrotor model to generate the closed-loop responses. In these simulations, we consider circular trajectories, with a range of target speeds of 0.8, 1.0 and 1.2 m/s and radii of 2, 3 and 4m. Each simulation run lasts 8 seconds. To test the adaptiveness of our framework, we consider two time instances during the simulation where we change the mass of the quadrotor mid-flight. At time = 2 seconds, the mass of the quadrotor is reduced by 50% and at time = 5 seconds, it is increased to 133% of the original mass. In simulation, we set data collection interval $t_{col} = 0.15s$ and neural network queue size $p = 3$. An implementation of our proposed framework can be found in this repository: https://github.com/TomJZ/Online-KNODE-MPC.

**Benchmarks.** We verify the performance of our proposed framework, KNODE-MPC-Online, by comparing against three benchmarks. First, we consider a standard nonlinear model predictive control (MPC) framework, where we use a discretized version of the nominal model $\tilde{f}(\mathbf{x}, \mathbf{u})$ as the prediction model in (6b). Comparing against this benchmark provides insights on the role of the neural network, in the presence of unknown residual dynamics. As a second benchmark, we consider the approach taken in [25], where the KNODE model is learned offline. This approach consists of two phases; data is first collected using a nominal controller and a KNODE model is trained using this collected data. The KNODE model is then deployed as the prediction model in (6b). We denote this approach as KNODE-MPC. By comparing our approach to KNODE-MPC, we show the effectiveness of online learning against residual dynamics and uncertainty that are possibly time-varying. In particular, to highlight the adaptive and generalization abilities of our approach, for the KNODE-MPC approach, we collect training data such that it only accounts for the first mass change at time = 2 seconds, but not the second mass change at time = 5 seconds. As a third benchmark, we consider a nonlinear geometric controller [34] to demonstrate the efficacy of the MPC algorithms.

## 4.3  Simulation Results

Table 1: Overall trajectory tracking mean squared errors (MSE) of our proposed framework against various baselines, across trajectories of different radii and speeds. For each speed and radius, the lowest MSEs are marked in bold.

| Radius [m] | 2.0 | | | 3.0 | | | 4.0 | | |
|---|---|---|---|---|---|---|---|---|---|
| Speed [m/s] | 0.8 | 1.0 | 1.2 | 0.8 | 1.0 | 1.2 | 0.8 | 1.0 | 1.2 |
| MPC | 0.0904 | 0.1280 | 0.1705 | 0.0949 | 0.1371 | 0.1861 | 0.0967 | 0.1412 | 0.1937 |
| KNODE-MPC [25] | 0.1222 | 0.1945 | 0.2555 | 0.1974 | 0.1769 | 0.2098 | 0.5303 | 0.4175 | 0.3418 |
| Geometric Control [34] | 0.2168 | 0.2572 | 0.3253 | 0.2067 | 0.2267 | 0.2606 | 0.2046 | 0.2194 | 0.2416 |
| KNODE-MPC-Online (ours) | **0.0660** | **0.1113** | **0.1678** | **0.0657** | **0.1043** | **0.1554** | **0.0709** | **0.1092** | **0.1571** |

Table 1 gives a comparison of our framework, KNODE-MPC-Online, against the benchmarks, MPC, KNODE-MPC and nonlinear geometric control. The plotted overall mean squared errors (MSE) are calculated by considering the difference between the time histories of the reference trajectory and the quadrotor position for each simulation run, and along each axis. They are calculated element-wise and indicate the overall trajectory tracking performance for each run. As shown in the table, KNODE-MPC-Online provides the best overall performance across different target speeds and radii. Notably, by considering the average across the runs, KNODE-MPC-Online outperforms MPC, KNODE-MPC and geometric control by 18.6%, 58.8% and 53.3% respectively. KNODE-MPC-Online performs well as it is able to account for the mass changes that occur during mid-flight. On the other hand, KNODE-MPC, with the KNODE model trained offline, is only able to account for the first mass change, but it is unable to account for the second change, as the effects of the second

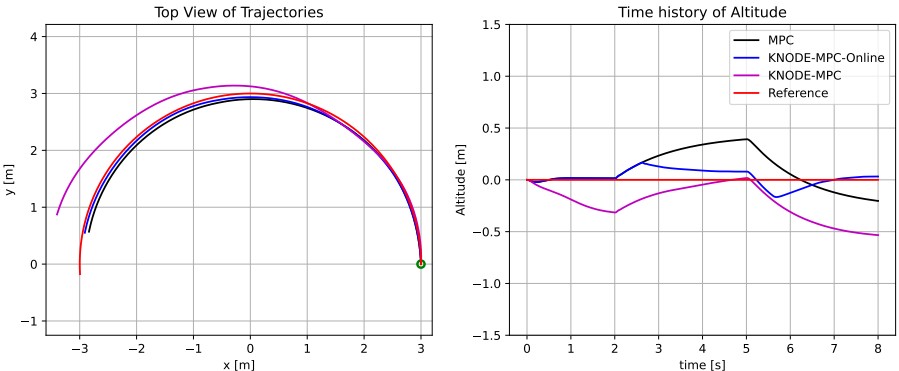

Figure 2: **Left panel:** Trajectory plot for MPC, KNODE-MPC and KNODE-MPC-Online (ours), from the top view. The reference trajectory is shown in red and the quadrotor position is initialized at $(x, y, z) = (3, 0, 0)$. **Right panel:** Time history of the altitude under the different frameworks.

mass change was not observed from its training data. Figure 2 illustrates the trajectories taken by the quadrotor when the MPC frameworks are deployed. The quadrotor is required to track a circular reference trajectory of radius 3m, shown in red. Evidently, from both the top view and time histories, the KNODE-MPC-Online framework allows the quadrotor to track the reference trajectory more closely, as compared to the two frameworks.

## 5  Physical Experiments

### 5.1  Experimental Setup

The open-source Crazyflie 2.1 quadrotor [35] is used as the platform for our physical experiments. The Crazyflie has a mass of 32g and has a size of 9 cm$^2$. A computer running on Intel i7 CPU is used as the base station and communication with the Crazyflie is established with the Crazyradio PA and at a nominal rate of 400 Hz. The software architecture is implemented using the CrazyROS package [36]. Position measurements of the quadrotor are obtained with a VICON motion capture system which communicates with the base station. Linear velocities are estimated from these positions, while accelerations and angular velocities are measured from the onboard accelerometers and gyroscope sensors. For these experiments, we implement a hierarchical control scheme where the nonlinear MPC scheme generates acceleration commands and pass them to a controller [34] to generate angle and thrust control commands, which are then sent to the lower-level controllers within the Crazyflie firmware. During each run, the Crazyflie is commanded at a target speed of 0.4m/s and to track a circular trajectory of radius 0.5m. We conduct 10 runs for each method, with a total of 40 runs. Similar to the mass changes done in simulations, we test the adaptiveness of the proposed framework by attaching an object of mass 1.3g (4.1% of the mass of Crazyflie) mid-flight. We compare our approach with the benchmarks described in Section 4.2. For the KNODE-MPC approach, training data is collected with the object attached, so that its effects can be accounted for in the model that is trained offline. In physical experiments, we set data collection interval $t_{col} = 2s$ and neural network queue size $p = 3$.

### 5.2  Results and Discussion

Results from the physical experiments are illustrated in Figure 3. The MSEs shown are defined in the same way, as described in Section 4.3. Statistics for each run are computed using 30 seconds of flight data, approximately starting from the time when the object is attached. As shown in the figure, our proposed approach, KNODE-MPC-Online, outperforms the benchmarks in terms of the overall MSE. In particular, the overall median MSE for KNODE-MPC-Online improves by 55.1%, 21.3% and 43.2%, as compared to nominal MPC, KNODE-MPC, and the geometric controller respectively. Furthermore, it is observed that results of KNODE-MPC-Online have a smaller variance as compared to the benchmarks. This implies that the approach is consistent in terms of control

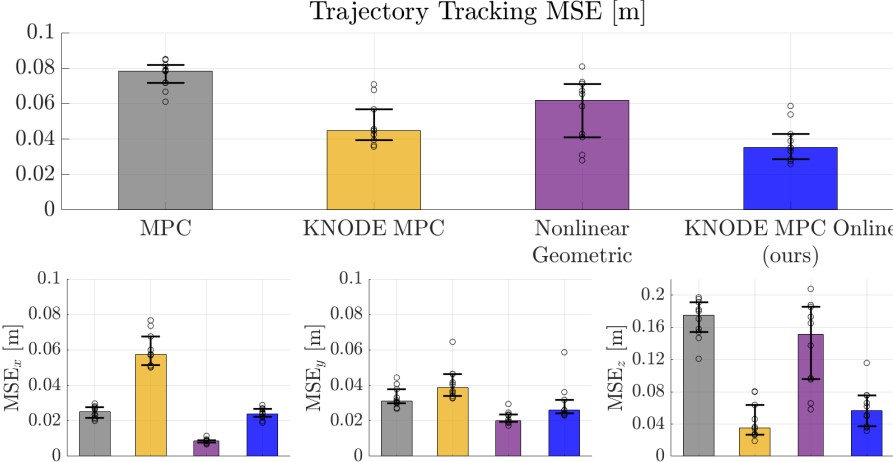

Figure 3: **Performance of KNODE-MPC-Online in physical experiments.** Statistics for trajectory tracking mean squared errors (MSE) for the benchmarks and our approach, KNODE-MPC-Online, are shown. The top of the bars denote the median, while the ends of the error bars represent the 25th and 75th percentiles. The top plot depicts the overall MSE and the bottom three plots shows the MSEs in separate axes.

performance across all runs. Since the nominal MPC and geometric control frameworks are unable to account for the mass perturbation mid-flight, the MSEs are observed to be generally larger, especially in the z axis. More notably, from the MSEs across each of the axes, we observe that the offline KNODE-MPC approach has larger errors in the x and y axes and smaller errors in the z axis. This suggests that the KNODE model trained offline is able to compensate for the mass perturbation induced in flight, since its effect is present in the training data. However, it is unable to compensate for errors in the x-y plane, which are likely not reflected in the training data. In contrast, the KNODE-MPC-Online approach achieves more consistent MSEs in all three axes and it is able to adapt to uncertainty during flight.

## 6    Conclusion

In this work, we propose a novel and sample-efficient framework, KNODE-MPC-Online, that learns the dynamics of a quadrotor robot in an online setting. We then apply the learned KNODE model in an MPC scheme and adaptively update the dynamic model during deployment. Results from simulations and real-world experiments show that the proposed framework allows the quadrotor to adapt and compensate for uncertainty and disturbances during flight and improves the closed-loop trajectory tracking performance. Future work includes applying this framework to other robotic applications where dynamic models can be learned to achieve enhanced control performance.

## 7    Limitations

A fundamental assumption of our framework is the continuous-time nature of system dynamics. This means our framework has limited applicability to stochastic systems. However, there has been variants of NODE which models stochastic differential equations. For future work, we hope to extend our algorithm to incorporate stochastic models to improve its applicability.

**Acknowledgments**

This work was supported by NSF IIS 1910308 and DSO National Laboratories, 12 Science Park Drive, Singapore 118225. The authors would also like to thank all reviewers and the area chair for their reviews and comments.

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
