# OpenReview forum: "Online Dynamics Learning for Predictive Control with an Application to Aerial Robots"
_robot-learning.org/CoRL/2022/Conference — CoRL 2022 Poster_

### Official Review · Reviewer_GTzB · 2022-07-26

**Originality:** Good
**Technical Quality:** Good
**Clarity Of Presentation:** Good
**Impact:** 4

**Recommendation:**

Weak Accept: I recommend accepting the paper, but will not argue for my recommendation if the majority of other reviewers have a different opinion.

**Summary:**

This paper presents an online learning framework in the context of model predictive control. The method includes a joint, parallel framework where the control scheme is updated based on new, incoming models; and the models are trained and updated based on the acquisition of new data in an online fashion. The models are based on the now-well-known neural ODE framework. The paper evaluates the method with a series of simulated quadrotor experiments w/r/t tracking error.

**Issues:**

Line 85: "First, it requires less data for training...". Less data than *what*? GPs? Standard neural networks? System identification?

I am a bit confused by equations (3) and (4). What is the time step in equation (3), i.e the difference between t_i and t_{i+1}? Is this a sort of design parameter, based on the dynamics of the system under consideration? Also, t_s is in T, i.e. any sampling time in a set. But how is T defined? Is it based on m, since we are averaging over a bunch of different start times? I conceptually understand what is going on here but the terms and nomenclature are slightly confusing.

How sensitive is this approach, generally speaking, to the numerical integration of equation (4). Numerical solvers / integrators are not always stable; I am curious to see if there are practical issues here.

Lines 113-118: I appreciate the discussion of t_{col} and the implications of longer or shorter collection intervals. However, it would be nice to see a discussion of how to characterize this tradeoff or identify the optimal tradeoff (perhaps this is coming later).

In equation (5) and the surrounding discussion, it is proposed to keep all (I think???) of the neural network-based models. This seems burdensome in terms of both time and memory. Can the authors comment on this choice? Why is it necessary to keep all of the preceding models? The authors make both an explicit claim and seem to have the implicit assumption that "more data means better models". I can think of counterexamples. Perhaps the authors feel the same way. The paper also provides no stability or asymptotic guarantees - i.e. that over time the model does indeed get better. Furthermore, the system needs to handle "jumps" in the model, which might cause locally "strange" behavior. i.e. the system experiences something it has not seen before and therefore the NN updating might not be smooth. In short: this type of behavior could be unsafe, even if the overall trend of the iteration is one of improvement.

Line 124: "to tackle this problem, we repeatedly add new neural networks in parallel" but it is unclear what is done with them. Does the selection matrix, M_phi, continue to adapt? Is this learned or learnable? Is this the mechanism that provides stability for learning?

The end of section 3.2 discusses the coupling between generating new data, training, and what model(s) the controller might be using In lines 130-133, it states that new data must be produce by an updated controller, and then refers to Alg. 2 Line 4. It is unclear how *waiting* (what the while loop does on line 4) implements this requirement.

It is still unclear why all the neural networks, f_theta, are needed when the recursion only depends on hat{f} and phi. Isn't it ok to discarbe and of the i^th or earlier neural networks? I am sorry I do not have a specific suggestion here, other than to say there is some confusion here, as well as some broader questions.

I appreciate the experimental setup and evaluation in section 5. However, the evaluation seems to be too "inward facing". As the authors acknowledge in the related works subsection, there is a lot going on in this area right now. And yet the evaluation includes MPC, KNODE-MPC, and the paper's online variant. As I mentioned earlier, I appreciate the authors' justification for choosing KNODE; however, it is still unclear whether this method would outperform other model-predictive RL (or control) methods that do or do not use machine learning. This question is especially concerning given that KNODE-MPC-Online seems to barely outperform vanilla MPC.

Lines 245-248: I agree that the continuous time nature of the dynamics could be a limitation. However, I was thinking more about mode switching and hybrid systems in this context, not necessarily the stochastic nature of the problem. Can't the uncertainty / stochastic systems still be modeled in a (discretized) variant of continuous dynamics? I feel like hybrid systems is the real problem here.

Lines 249-252, the authors acknowledge one of my earlier concerns. One thing that is unclear is how substantial this growth in computational load is: is it exponential in the amount of time under experiment? And then, if we do network pruning or reduced order modeling, do we reduce the accuracy of the models in such a way to render the method ineffective? Furthermore, I had an earlier comment about the tradeoffs in increasing or decreasing collection time. Perhaps I missed this, but a more rigorous assessment of this tradeoff would be nice.

**Quality Of The Limitations Section:**

Additional details required

**Reviewer Expertise:**

4: The reviewer is confident but not absolutely certain that the evaluation is correct

**Robotics Focus:**

Highly relevant to robotics but no hardware experiments

**Strengths And Weaknesses:**

Strengths
- The justification for KNODE presented in lines 82-92 is appreciated.
- The limitations section discusses several of my concerns, indicating that the the authors seem to anticipate several questions about the work that might arise

Weaknesses
- Maintaining a potentially large number of neural networks seems burdensome and possibly unnecessary
- There is a lack of a discussion or theory w/r/t convergence guarantees. The paper seems to imply that "more data is better" but this is not always the case, both in terms of the data that is generated itself and/or in terms of the algorithms that are using these data

**Summary Of Recommendation:**

The paper presents an interesting idea and a relevant topic for robotics (and other applications). The paper would be significantly improved with a broader set of evaluation baselines and/or more exposition on theoretical guarantees in the absence of more rigorous experimental (simulation) evalutions.

---

> ### Author Response · Authors · 2022-08-17
> **Clarified and addressed concerns for the framework**
>
> We thank the reviewer for the comment on the burdensome use of neural networks. We have revised our algorithm to achieve constant memory and computational cost by using forgetting factors on the output of neural networks, and by maintaining a fixed number of neural networks. The details of our revised algorithm can be found in our reply to Reviewer sQ7A.
>
> The authors agree with the reviewer that “more data is better” is not always the case. In fact, we think there are two reasons for this not being entirely correct. First, having more data increases the training time, and therefore may reduce the adaptiveness of real-time algorithms. Second, as the true dynamics of a robot changes during deployment, data collected over long periods of time may not come from the same distribution anymore, and hence having “more data” may not give an accurate representation of the robot dynamics. With our revised algorithm, we aim to address the second problem by forgetting older neural networks. The older neural networks may contain information about the robot dynamics that may have become obsolete. The FIFO queue of neural networks therefore represents the most up-to-date information about the robot dynamics. To address the first problem, we do a hyperparameter search on the collection duration $t_{col}$. The results of the hyperparameter search are included in our response to the Area Chair EkgR. We select the best data collection duration by considering the tradeoff between performance and adaptiveness of the algorithm.
>
> In Line 85, we mean less data than when no knowledge/physical prior is used. In other words, when a blackbox neural network is used to represent the entire dynamics model. The coupling of the neural networks and nominal model is described in detail in our response to reviewer YuBR.
>
> There are two different notions of “time steps” in equations 3 and 4. Both equations are written with respect to a continuous-time model, and therefore there is no notion of “time step” in $\hat{f}$, the model itself. The integrals in these two equations will be evaluated using numerical solvers, which are then associated with time steps. In other words, the solvers will discretize these two equations into discrete time steps. There is a second notion of time step associated with sampling. Although the dynamics model is inherently continuous, the training data is always sampled from discrete time steps. Hence, $t_i$ and $t_{i+1}$ represent the discrete time steps in the sampled data. $t_s$ represents any sampling time that exists within the set of time stamps. The use of the Dirac delta function is to force the MSE computed at these discrete time steps to come out of the integral (Otherwise they will be zero). Note that the two notions of time steps do not necessarily have to be the same. For example, if two adjacent time steps in the training data are 0.0s and 0.3s, we can set a numerical solver to take three steps to go from 0.0 to 0.3, with each step being 0.1s. This formulation leverages one of the advantages of KNODE, which is that it can learn from irregularly sampled data.
>
> Stability of numerical solvers is widely discussed in dynamics learning. The issue arises especially for challenging dynamics such as stiff systems. The quadcopter model is not stiff and therefore we have not encountered any numerical issue with the solver. Additionally, we use one-step-ahead predictions to compute the loss function, where the step size is small, and therefore even if the model is chaotic/unstable, the training will not diverge [3].
>
> We have included the hyperparameter search and selection in our response to the Area Chair. And for theoretical guarantees, we will leave it for future work. As for the waiting, while the training module waits, the data collection module is actively writing new data to be stored for later training.
>
> We appreciate the reviewer for acknowledging the experimental setup and evaluation in Section 5. The baselines are selected to highlight the features of our proposed framework, KNODE-MPC-Online, as described in lines 175-185. To give a better idea on how the methods perform against other control methods, we include a nonlinear geometric controller [1], which is a non-MPC controller that does not include any learning components. Results for the nonlinear geometric controller, our revised algorithm and existing methods can be found in our reply to Reviewer sQ7A. Additionally, we would like to refer the reviewer to [2], where a comparison between KNODE-MPC and GP-MPC is made. We would also like to highlight that KNODE-MPC-Online outperforms vanilla MPC quite substantially in physical experiments which are described in Section 5 of our manuscript and this can be observed from Fig. 4, as well as from the second table in our reply to Reviewer sQ7A.
>
> We agree with the reviewer that it’s unclear how KNODE can model hybrid systems. We will be sure to include this in the limitation section in our revised manuscript.

---

### Official Review · Reviewer_sQ7A · 2022-07-28

**Originality:** Fair
**Technical Quality:** Fair
**Clarity Of Presentation:** Fair
**Impact:** 2

**Recommendation:**

Strong Reject: I recommend rejecting the paper and will argue for my recommendation even if other reviewers hold a different opinion.

**Summary:**

This paper developed a framework for online updating of the dynamics modeled by KNODE for MPC.
This ensured adaptability to changes in the environment, and even when unknown weights were actually attached to the drone, that change was reflected appropriately in the dynamics to achieve optimal control.

**Issues:**

I would like to see a proper survey of previous studies, as online methods for updating dynamics are not uncommon.

The specific MPC solver was unknown unless I am missing something.
Some solvers are computationally very expensive, and optimization should not be possible on time for methods such as the proposed method, where the computational cost of the dynamics explodes.
With this in mind, please clarify the solver and discuss the realtimeness of the optimization.

**Quality Of The Limitations Section:**

Limitations are not well addressed

**Reviewer Expertise:**

3: The reviewer is fairly confident that the evaluation is correct

**Robotics Focus:**

Highly relevant to robotics but no hardware experiments

**Strengths And Weaknesses:**

Strengths:
- The online adjustment of dynamics enhanced adaptability to the environment.
- The proposed method introduced a trick to avoid forgetting the past, which is an important point in online learning.


Weaknesses:
- Online dynamics learning itself is not new.
- In avoiding past forgetting, the append-type method diverges in both memory and computational cost.
- The adaptability of the proposed method is not well demonstrated with only one example.

**Summary Of Recommendation:**

In the context of model-based RL, it is not uncommon to update dynamics online, which is not consistent with the claims of this paper.
Also, if online learning is anticipated, a structure such as the proposed method, which ultimately requires infinite memory and computational cost, is not appropriate.
As mentioned in the Limitations section, I cannot find the value of this method unless the proposed method incorporates tricks to keep the computational cost constant.

---

> ### Author Response · Authors · 2022-08-17
> **Revised algorithm to have constant memory and computational cost**
>
> We thank the reviewer for noting our insufficient related works on online model learning. To further illustrate the difference between our method and existing online dynamics learning literature, we will include a paragraph in our related works section, found in the PDF attached.
>
> We appreciate the reviewer’s comment on memory and computational cost. The authors also recognize this as a major weakness of the proposed method. We have therefore revised our algorithm by maintaining only a fixed number of NNs during online learning, and using forgetting factors on the outputs of the NNs. This addresses the problems with both the diverging memory and computational costs. Specifically, given $p$, the maximum queue size for NNs, new NNs are pushed to the queue until the maximum queue size is reached. Afterwards, addition of new NNs will pop old NNs in a first-in-first-out (FIFO) manner. For weighting the NNs, we modify the first line of equation (5) to be $$\hat{f}^{(i+1)} = M_{\psi_{(i+1)}}(\hat{f}^{(i)}, e^{i+1-p}f_{\theta_{(i+1)}})\ \text{for}\ i< p$$. This will give a weight of 1 for the output of the newest NN, and exponentially decaying weights for older NNs. With these improvements, the memory and computational cost remain constant regardless of deployment duration. In addition to the collection interval, we also have the hyperparameter $p$ that needs to be set. In our previous implementation, to balance the tradeoff between adaptiveness and computational cost, we used $t_{col} = 0.5$. Our improved algorithm frees us from this constraint, and allows us to use more optimal $t_{col}$. The resulting tracking MSEs are better than all of the approaches, including our previous algorithm. Results are shown in Table I in the attached PDF (best MSE in bold).
>
> We have also conducted additional physical experiments to verify our revised framework. As we can expect, given the fixed size of the NN pool, it is observed that KNODE-MPC-Online (new) has a fixed computational cost. The results for our revised framework, with the existing results, are summarized in Table II. The best MSEs in terms of the 25th, 75th percentiles and the median, are highlighted in bold.
>
> We thank the reviewer for the comments on the MPC solver. In the initial manuscript, a description of the solver for the nonlinear MPC problem is given in lines 167-168. For better clarity, we will move this description to Section 3.3. Additionally, we will add more details of the solver implementation in the same section. The following paragraph will be included in Section 3.3:
>
> “The optimization problem (6) is formulated and implemented using the Opti-stack class within the CasADi library [1]. An interior-point method IPOPT [2] within the library is used to solve the problem. The solver is warm-started at each time step by providing it with an initial guess of the solution, which is based on the optimal solution, both primal and dual variables, obtained from the previous time step.”
>
> Following up on the revised description, we would like to give an illustration on how warm-starting is useful in our implementation by considering one of the test cases in the simulations. Consider the case where the quadrotor is commanded to track a circular trajectory of radius of 3m and with a speed of 1m/s. The quadrotor is controlled using a standard MPC framework, where there are no learning components. The simulation is run for 5 times, with and without warm-starting. Without warm-starting, it takes an average of 64.3 seconds, and with warm-starting, it takes an average of 36.7 seconds. That gives computational savings of more than 40%. However, based on the current implementation, we are unable to apply the framework directly to a full-dimensional quadrotor in real-time. In the physical experiments, we implement a hierarchical control scheme where the nonlinear MPC generates acceleration commands and pass them to a controller [3] to generate angle and thrust control commands, which are sent to the lower-level controllers within the Crazyflie firmware. This enables us to verify the proposed framework in real-time experiments. We will add these details about the implementation in the physical experiments in our revised manuscript.

---

> > ### Comment · Reviewer_sQ7A · 2022-08-27
> > **Thank you for your response**
> >
> > Thank you for the additional experiments. However, FIFO is a very naive implementation, and I think it is not a good solution, as it will end up forgetting everything except the most recent. It would be worthwhile to do a thorough study of continual learning and update your method to properly sort out the information that should be forgotten from the information that should not be forgotten.

---

> > > ### Author Response · Authors · 2022-08-27
> > > **Justification for FIFO and exponential forgetting factors**
> > >
> > > We thank the reviewer for the follow-up comment. We do not agree with the reviewer that FIFO with exponential forgetting factors is a  “very naive implementation,” we instead think it is a natural and simple solution that works well for our application. This solution solves the problem with infinite memory and computational costs, which were brought up by a number of reviewers. More importantly, this solution has been validated through our additional experiments. As demonstrated in the additional experiments, including the experiments with the control uncertainty as suggested by Reviewer Jxoq, this solution works better than all baselines, including our previous method with no forgetting. Our new results confirm an important observation on how online learning of residual dynamics may be done in a simple and straightforward manner. More detailed justification of the choice of FIFO implementation can be found in our second comment to reviewer Jxoq. And since our new implementation beats our previous method where no forgetting is done, it validates the observation that **having more recent data is better**. It is true that as the queue of NNs updates, old information will eventually be forgotten, but note that the NNs only learn the residual dynamics, while the knowledge (nominal model) still retains the majority of information about the robot dynamics. Hence, the reviewer’s comment that our NNs “end up forgetting everything except the most recent” may not be a bad outcome after all. This is verified by our simulation and physical experiments results.
> > >
> > > While the authors agree with the reviewer that there are a variety of continual learning methods, a primary objective of this work is on online learning, and not continual learning. In fact, the tasks considered in continual learning are different from the application in this work. Continual learning addresses the problem of catastrophic forgetting over long time spans [1,2,3,4], which contradicts the idea of "having more recent data is better". We show that online learning of neural networks can adaptively improve MPC performance in a real-time fashion. This is achieved using small batches of data, with our proposed iterative update scheme. In our revised related works section, it can be seen that existing online learning approaches with MPC either use model parameterizations other than neural networks, or require prior data for offline training before deployment. In contrast, we use neural networks and avoid offline training, which are improvements to these methods.
> > >
> > > ---
> > >
> > > [1] Z. Chen and B. Liu, “Lifelong machine learning,” Synthesis Lectures Artif. Intell. Mach. Learn., vol. 12, no. 3, pp. 1–207, 2018.
> > >
> > > [2] G. I. Parisi, R. Kemker, J. L. Part, C. Kanan, and S. Wermter, “Continual lifelong learning with neural networks: A review,” Neural Netw., vol. 113, pp. 54–71 2019.
> > >
> > > [3] H. Shin, J. K. Lee, J. Kim, and J. Kim, “Continual learning with deep generative replay,” in Proc. Int. Conf. Neural Inf. Process. Syst., 2017, pp. 2990–2999.
> > >
> > > [4] Vijayan, M., Sridhar, S.S. (2021). Continual Learning for Classification Problems: A Survey. In: Krishnamurthy, V., Jaganathan, S., Rajaram, K., Shunmuganathan, S. (eds) Computational Intelligence in Data Science. ICCIDS 2021. IFIP Advances in Information and Communication Technology, vol 611. Springer, Cham.

---

### Official Review · Reviewer_Jxoq · 2022-07-29

**Originality:** Good
**Technical Quality:** Good
**Clarity Of Presentation:** Excellent
**Impact:** 3

**Recommendation:**

Strong Accept: I recommend accepting the paper and will argue for my recommendation even if other reviewers hold a different opinion.

**Summary:**

The major contribution of this paper is to augment the existing work [20] to be implemented for online training. For the learning of dynamics to execute online, neural networks are stacked with weighted outputs to progressively capture the new features in the vehicle dynamics. Experiments are conducted to show the capability of the proposed method to adapt to changing dynamics of a quadrotor.

**Issues:**

A couple of minor issues are listed below in addition to the major ones summarized in the “Strengths And Weaknesses” section:
1. In line 49, towards the end, there is a typo ‘hysical.’
2. In line 81, a period is missing between [11] and knowledge-based
3. In figure 2, given that the heatmap does not show a clear tendency in the 3x3 grids, it might be more straightforward to turn the MSE into a table with numbers.
4. In figure 3, the plot on the right can be more informative if the horizontal axis is time instead of y position (to show the altitude holding capability).
5. Please remove the last paragraph in Section 7 as it is not related to the paper's focus.


**Quality Of The Limitations Section:**

Limitations are addressed clearly

**Reviewer Expertise:**

5: The reviewer is absolutely certain that the evaluation is correct and very familiar with the relevant literature

**Robotics Focus:**

Sufficient demonstration on hardware

**Strengths And Weaknesses:**

The major strength of the paper is to design an iterative update scheme to incorporate newly learned NN to describe the dynamical model from online data better. The more precise dynamics will yield better knowledge in the MPC controller and hence improve tracking performance subject to changes in the dynamics.

The major weakness of the paper is the lack of description/explanation of the selection mechanism of the stacked NN. Specifically, the authors mentioned the selection matrix M_\psi, which “couples the neural network with knowledge.” However, there’s no description of how the selection matrix is formed or designed. The ambiguity introduces further questions to the proposed framework. For example, what if a newly introduced NN contains the information captured by a previously stacked NN? This question tailors more towards the efficiency of stacking NN’s where it is possible that the NN’s contain repeated knowledge.

Another weakness is that the paper frequently refers to the adaptiveness of the proposed framework. Yet the literature review covers no results from the adaptive control community. However, there are recent adaptive control designs that work fairly well for quadrotor control. The authors should include it in the literature review and cite related works, e.g.,
[1] Kostadinov, D. and Scaramuzza, D., 2020, August. Online weight-adaptive nonlinear model predictive control. In 2020 IEEE/RSJ International Conference on Intelligent Robots and Systems (IROS) (pp. 1180-1185). IEEE.
[2] Hanover, D., Foehn, P., Sun, S., Kaufmann, E. and Scaramuzza, D., 2021. Performance, precision, and payloads: Adaptive nonlinear mpc for quadrotors. IEEE Robotics and Automation Letters, 7(2), pp.690-697.
[3] Joshi, G., Virdi, J. and Chowdhary, G., 2020. Asynchronous deep model reference adaptive control. arXiv preprint arXiv:2011.02920.
[4] Wu, Z., Cheng, S., Ackerman, K.A., Gahlawat, A., Lakshmanan, A., Zhao, P. and Hovakimyan, N., 2022, May. L 1 Adaptive Augmentation for Geometric Tracking Control of Quadrotors. In 2022 International Conference on Robotics and Automation (ICRA) (pp. 1329-1336). IEEE.


**Summary Of Recommendation:**

I recommend this paper to be accepted, given that the authors can clearly explain how the selection matrix M_\psi is designed and include the references of adaptive control for quadrotors.

---

> ### Author Response · Authors · 2022-08-17
> **Clarified details of the method and added related works on adaptive control**
>
> We thank the reviewer for the comment on the structure of the stacked NN, which was also pointed out by Reviewer YuBR. In our reply to Reviewer YuBR, we have described the structure of the M matrix and how the outputs from the NNs are coupled. We will explain this more clearly in our revised manuscript. We also agree that repeatedly stacking NNs may incur unnecessary memory and computational cost, especially when new NNs contain repeated knowledge. We have revised our algorithm, which is described in detail in our response to Reviewer sQ7A. In the revised algorithm, only a fixed number of NNs are used. The outputs of the NNs are weighted in such a way that the newest NN has a weight of 1 and the weights exponentially decay for older NNs. The oldest NN (and therefore with the smallest weight) will be discarded when a new NN is added to maintain the fixed number of NNs. The revised algorithm is an improved solution to the online learning problem, because as the dynamics of a robot changes during deployment, the older information may become obsolete, and therefore constantly “flushing” the collection of NNs helps to maintain the most up-to-date dynamics model in the controller.
>
> We thank the reviewer for the comment on related works. We have looked at the recommended references, as well as other relevant references related to adaptive control. The references we provide in our initial manuscript are those that we think are the most relevant to our framework – learning the dynamics model within an MPC framework. To improve the related works section, we will add the following paragraph, which includes references to adaptive control:
>
> “There are a number of works in the literature that promote adaptiveness within a MPC framework. In [1], the authors propose updating the state and control weights in the MPC optimization problem in an online setting. The authors in [2] propose an adaptive cruise control scheme that includes a real-time weight tuning strategy, where the weights in the optimization problem are adjusted based on various operating conditions.  In our work, instead of optimizing for the cost weights or other parts of the objective function during deployment, we use deep learning to compensate for the uncertain dynamics that are not accounted for in the dynamics constraints within the MPC framework. In [3], a nonlinear MPC scheme augmented with a L1 adaptive component is proposed. The errors between the state estimator and L1 observer are incorporated into the control scheme through an adaptation law. In a similar vein, the authors in [4] proposed a linear MPC scheme combined with a L1 adaptive module for quadrotor trajectory tracking. Joshi et al. [5] use a learning rule to update weights of the outermost layer of the neural network. This network is then used in the adaptive term in the proposed controller for a quadrotor. In [6], a L1 adaptive control augmentation is incorporated into a geometric controller. In contrast to the above works, instead of augmenting existing controllers with an adaptive module, our proposed framework directly updates the dynamic constraints within the model predictive controller and the control inputs are obtained by solving the optimization problem.”
>
> We thank the reviewer for pointing out the typos and suggesting possible modifications in the manuscript. We shall amend the typos, convert Fig. 2 into a table, change the x axis of the right subplot of Fig. 3 to indicate time in the revised manuscript. For the table, we will also include results for the revised scheme as well as another non-MPC baseline, a nonlinear geometric controller. This table that replaces Fig. 2 can be found in our reply to Reviewer sQ7A. We will also remove the last paragraph in Section 7.
>
> —
>
> [1] Kostadinov, D. and Scaramuzza, D., 2020, August. Online weight-adaptive nonlinear model predictive control. In 2020 IEEE/RSJ International Conference on Intelligent Robots and Systems (IROS) (pp. 1180-1185). IEEE.
>
> [2] R. C. Zhao, P. K. Wong, Z. C. Xie, and J. Zhao. Real-time weighted multi-objective model predictive controller for adaptive cruise control systems. International Journal of Automotive Technology, 18:1976– 3832, 2017.
>
> [3] Hanover, D., Foehn, P., Sun, S., Kaufmann, E. and Scaramuzza, D., 2021. Performance, precision, and payloads: Adaptive nonlinear mpc for quadrotors. IEEE Robotics and Automation Letters, 7(2), pp.690-697.
>
> [4] Pereida, K., & Schoellig, A. P., 2018, October. Adaptive model predictive control for high-accuracy trajectory tracking in changing conditions. In 2018 IEEE/RSJ IROS (pp. 7831-7837). IEEE.
>
> [5] Joshi, G., Virdi, J. and Chowdhary, G., 2020. Asynchronous deep model reference adaptive control. arXiv preprint arXiv:2011.02920.
>
> [6] Wu, Z., Cheng, S., Ackerman, K.A., Gahlawat, A., Lakshmanan, A., Zhao, P. and Hovakimyan, N., 2022, May. L 1 Adaptive Augmentation for Geometric Tracking Control of Quadrotors. In 2022 ICRA (pp. 1329-1336). IEEE.

---

> > ### Comment · Reviewer_Jxoq · 2022-08-25
> > **response to the authors**
> >
> > I appreciate the changes made by the authors to clarify the paper. One follow-up question regarding the exponential decaying factor is the following: How the weighted sum of outputs from different NNs can work? Each NN is learned to reduce the residual error. Thus, it's unclear what it means when applying weights to the most recent p number of NN outputs. The authors should justify it in addition to showing improved experimental results, which only account for empirical evaluation. It seems more of a working heuristic which should be accompanied by some analysis. One simple idea is to apply an injected uncertainty into the control channel and see how the proposed method can cancel out this uncertainty. The method has been applied in [6] in the reference list above.

---

> > > ### Author Response · Authors · 2022-08-26
> > > **Justified exponential decaying weights and additional experiments on the effect of control uncertainty**
> > >
> > > We thank the reviewer for the follow-up question on the exponential decaying factors on NN weights. Our previous algorithm would be applicable if system changes only happen once, which means that all NNs that are added to the ensemble are bridging the same set of residual dynamics. However, demonstrated by our simulations, the system can change more than once, and the uncertainty could be time-varying as illustrated in [6]. In this case, the previous algorithm would be less relevant. Hence, having decaying weights can be justified by the assumption that the residual dynamics constantly change as a result of constantly changing system dynamics. In other words, the residual dynamics that have been captured earlier may not be the most recent residual dynamics anymore, and the NNs that capture the old dynamics should fade out and eventually be discarded.
> > >
> > > We appreciate and thank the reviewer for suggesting the simple idea of injecting uncertainty in the control channel. With reference to the formulation in [6], we inject a time-varying control input uncertainty into the dynamics of the quadrotor in our simulation setup. The control input uncertainty is injected at the start of the simulation. We also include a scale factor to account for the difference in mass and thrust between the quadrotor systems considered in [6] and the one used in our paper. We conduct two sets of experiments. The first set considers only the control input uncertainty, without the system changes proposed in our initial manuscript. The second set considers the system changes, on top of the control input uncertainty. Results are given in Tables I and II below. The effect of disturbance is the difference between the MSEs under the cases with and without uncertainty. Overall (looking across both sets of experiments), it is observed that under our revised scheme KNODE-MPC-Online (new), the effect of disturbances is the smallest among the baselines.
> > >
> > > **Table I: Effect of disturbance on performance (without system changes)**
> > > |         Method         | No uncertainty (factor = 0.0) | With thrust_disturb (factor = 0.5), no roll_moment_disturb |     | With thrust_disturb (factor = 0.75), no roll_moment_disturb |     | With roll_moment_disturb (factor = 0.1), no thrust_disturb  |     |
> > > |:----------------------:|-------------------------------|------------------------------------------------------------|-----------------------|-------------------------------------------------------------|-----------------------|-------------------------------------------------------------|-----------------------|
> > > |      | MSE         | MSE      | Effect of disturbance | MSE       | Effect of disturbance | MSE       | Effect of disturbance |
> > > | MPC  | 0.124109    | 0.129256   | 0.005147     | 0.137095  | 0.012986     | 0.220802  | 0.096692     |
> > > | KNODE-MPC     | 0.159780129          | 0.165602   | 0.005822     | 0.176078  | 0.016298     | 0.337138  | 0.177358     |
> > > | KNODE-MPC Online       | 0.114518    | 0.117665   | **0.003147**     | 0.124097  | 0.009579     | 0.165482  | 0.050964     |
> > > | KNODE-MPC Online (new) | 0.099597    | 0.103365   | 0.003768     | 0.109144  | **0.009547**     | 0.139047  | **0.039450**     |
> > > | Geometric Control      | 0.050340    | 0.157980   | 0.107640     | 0.499856  | 0.449516     | 35.595970   | 35.545630    |
> > >
> > >
> > >
> > > **Table II: Effect of disturbance on performance (with system changes, same mass changes as our simulation setup)**
> > > |         Method         | No uncertainty (factor = 0.0) | With thrust_disturb (factor = 0.5), no roll_moment_disturb |     | With thrust_disturb (factor = 0.75), no roll_moment_disturb |     | With roll_moment_disturb (factor = 0.1), no thrust_disturb  |     |
> > > |:----------------------:|-------------------------------|------------------------------------------------------------|-----------------------|-------------------------------------------------------------|-----------------------|-------------------------------------------------------------|-----------------------|
> > > |      | MSE         | MSE      | Effect of disturbance | MSE       | Effect of disturbance | MSE       | Effect of disturbance |
> > > | MPC  | 0.137060601 | 0.142222   | 0.005161     | 0.151571  | 0.014511     | 0.233119  | 0.096059     |
> > > | KNODE-MPC     |  0.17686865 | 0.183685   | 0.006816     | 0.196459  | 0.019590     | 0.357427  | 0.180558     |
> > > | KNODE-MPC Online       | 0.124161549 | 0.129071   | 0.004909     | 0.138006  | 0.013844     | 0.175244  | 0.051082     |
> > > | KNODE-MPC Online (new) | 0.104339864 | 0.108452   | **0.004112**     | 0.1181468   | **0.013807**     | 0.153081  | **0.048741**     |
> > > | Geometric Control      | 0.226657225 | 0.233090   | 0.006433     | 0.927247  | 0.700590     | 65.586820   | 65.360163    |

---

### Official Review · Reviewer_YuBR · 2022-08-05

**Originality:** Fair
**Technical Quality:** Good
**Clarity Of Presentation:** Fair
**Impact:** 3

**Recommendation:**

Weak Accept: I recommend accepting the paper, but will not argue for my recommendation if the majority of other reviewers have a different opinion.

**Summary:**

In this paper, the authors propose KNODE-MPC-Online, an MPC that relies on Knowledge Based Neural Ordinary Differential Equations as dynamic models and updates the model online using neural networks. Experiments are conducted both in simulation and on a real robot.

**Issues:**

* Clarity
* Some more experiments missing (minor)

**Quality Of The Limitations Section:**

Limitations are addressed clearly

**Reviewer Expertise:**

4: The reviewer is confident but not absolutely certain that the evaluation is correct

**Robotics Focus:**

Sufficient demonstration on hardware

**Strengths And Weaknesses:**

Pros:
* The paper is well written and well structured
* Related work is sufficiently addressed
* Real world experiments are conducted and limitations are discussed
* The approach is interesting and pseudocode is provided in the form of algorithm

Cons:
* Some clarifications should be provided about the approach which is currently not reproducible
according to how it is described:
  - How do you compute the M matrix?
  - It is unclear how you use learned model in the MPC (Although there is a section about that in the paper). Specifically, given you learn several models, how do you 'choose' (or mix) them in order to obtain your prediction?
* It would be interesting to compare against other learning methods (even if not online) with MPC
to have a better sense of how it performs

**Summary Of Recommendation:**

The paper is well written and well structured, but it lacks some clarity in the approach which make it currently difficult to re-implement.

Although the paper is somewhat incremental, some clarity would make it much better and potentially useful. Experiments are interesting,
but it would have been nice to have a larger set of baselines to compare against that make use of learning with MPC (even if not online).

---

> ### Author Response · Authors · 2022-08-17
> **Explanation of the framework**
>
> We thank the reviewer for the comment on the clarity of the approach. In our work, we assume that the NNs are additive, and therefore the M matrices are formed by stacking two n by n identity matrices (with n being the output dimension of the MPC model). The output of each of the NNs and the nominal model are sequentially coupled with the M matrices. Or to put it simply, their outputs are added together. We will be sure to make this clearer in Section 3.2 in our revised manuscript. Please note that we have revised our algorithm to resolve the issue with increasing memory and computational cost. The revised algorithm is described in our response to Reviewer sQ7A. Furthermore, we have plans to make the implementation of our proposed framework available. This will make re-implementation more convenient and results reproducible.
>
> We appreciate the reviewer’s recommendation on the comparison against other learning methods. We would like to refer the reviewer to [1] for a comparison between KNODE-MPC and GP-MPC. Given this comparison in [1] as background, it has been shown in our results that our proposed framework is an improvement over KNODE-MPC, and indirectly over GP-MPC. Additionally, we include another non-MPC baseline, a nonlinear geometric controller [2], in which we implement in both simulations and physical experiments and compare it with the existing baseline methods. The simulation and physical experimental results for the geometric controller and our revised framework, together with the existing results, can be found in our reply to Reviewer sQ7A.
>
> —
>
> [1] K. Y. Chee, T. Z. Jiahao, and M. A. Hsieh. KNODE-MPC: A knowledge-based data-driven predictive control framework for aerial robots. IEEE Robotics and Automation Letters, 7(2): 2819–2826, 2022.
>
> [2] D. Mellinger and V. Kumar, “Minimum snap trajectory generation and control for quadrotors,” in 2011 IEEE Int. Conf. on Robot. and Automat., 2011, pp. 2520–2525.

---

### Meta-Review · Area_Chair_EkgR · 2022-08-05

**Recommendation:** Accept (Poster)
**Confidence:** 2

**Metareview:**

### Strengths
- interesting combination of MPC and KNODE (knowledge-based neural differential equations)
- hardware experiments with the crazyflie (with an external computer)

### Weaknesses
- no detail about the solver and, more importantly, about the overall computing cost of the approach, especially for long experiment (more data)
- not many baselines (eg non-MPC baselines)
- some "ablations" might be useful to understand the contribution of each component
- no source code is provided
- the related work section can be improved
- no theoretical guarantees
- statistics: how many times was the *training* algorithm run with different seeds?

### Post-rebuttal update
I would like to thank the authors for their effort to improve their paper.

I agree with one of the reviewers that the technique itself is not truly original (MPC + model learning is not novel). In addition, the statistics are too weak for empirical results: a single experiment with only 5 seeds is not enough to conclude that the training works reliably (the training algorithm might have been lucky... or the authors might have chosen their network well). However, I do think that combining learning and MPC is a very interesting and promising topic and this paper is an interesting contribution in that direction.

---

> ### Author Response · Authors · 2022-08-17
> **Addressing the concerns pertinent to the framework**
>
> We thank the Area Chair EkgR for the meta-review, comments, suggestions and recommendations.
>
> We have provided more details on the solver and the overall computing cost of our approach in our replies to Reviewers sQ7A and GTzB.
>
> We have added a nonlinear geometric controller [1] as an additional non-MPC baseline. Furthermore, we have revised our framework, which provides better accuracy in terms of MSE, with a fixed computational cost. This is described in detail in our replies to the four reviewers. The previous framework also serves as a baseline, in which we can compare it against the current algorithm, which can also be interpreted as a form of an ablation study. We have consolidated the simulation and physical experimental results of the geometric controller with those of our revised framework, denoted as KNODE-MPC-Online (new), in the form of tables, which can be found in our reply to Reviewer sQ7A.
>
> We have conducted “ablation”/sensitivity studies with respect to the different components in our revised framework, namely (1) with or without L2 regularization in training (implemented in the form of weight decay of the Adam optimizer), and (2) hyperparameter search on length of training data $t_{col}$, and size of NN pool $p$. These analyses will be included in our revised manuscript. Specifically, having no L2 regularization not only degrades the controller performance but may also slow down the nonlinear solver. For example, for a circular trajectory with radius 3, a speed of 1, $t_{col}=0.15$, and an NN pool size of 3, the overall trajectory MSE when not using L2 regularization is 5.195, while for with L2 regulation is 0.1043 and the simulation run was completed 40% faster. This difference is significant. Results for the hyperparameter search on $t_{col}$ are shown in the following table (with NN pool size = 4, running with circular trajectory with radius=3, speed=1),
>
> | $t_{col}$ | 0.02   | 0.05   | 0.1    | 0.15   | 0.2    | 0.3    | 0.4    | 0.5    |
> |-----------|--------|--------|--------|--------|--------|--------|--------|--------|
> | mse       | 0.1241 | 0.1200 | 0.1050 | 0.1045 | 0.1216 | 0.1259 | 0.1385 | 0.1365 |
>
> where we observe that for very small data collection durations, the training data is not sufficient and therefore the performance degrades. For large $t_{col}$, as we discussed in the paper, the adaptiveness will suffer. Hence we select $t_{col} = 0.15$ in our experiments as a tradeoff between performance and adaptiveness.
>
> Similarly, the results for the hyperparameter search on the size of the NN pool is summarized in the table below:
> | pool size, $p$ | 1      | 2      | 3      | 4      | 5      |
> |-------------|--------|--------|--------|--------|--------|
> | mse       | 0.1046 | 0.1059 | 0.1043 | 0.1045 | 0.1043 |
>
> where we select $p=3$. For larger pool size, the computational cost (for both training and MPC optimization) will increase as a result of increasing model complexity.
>
> We will open source our code, which enables re-implementation and results to be reproducible.
>
> We are adding two paragraphs to improve the related works section; one will cover works related to adaptive control, as pointed out by Reviewer Jxoq and the other will cover works related to online dynamics learning, as pointed out by Reviewer sQ7A. We have listed the additional paragraphs and references in our replies to Reviewers Jxoq and sQ7A.
>
> We thank the Area Chair EkgR for the comment on the lack of theoretical guarantees. We agree that the proposed framework does not have any form of theoretical guarantees and results are empirical. We plan to provide some theoretical basis for our proposed framework in our future work.
>
> To study the effect of different seeds in the training component of our revised framework, we consider one of the test scenarios in simulation (radius=3m, speed=1m/s) and note the results across 5 different seeds. The mean and median MSEs of these 5 runs are at 0.1269 and 0.1246 respectively, which are comparable to that given in the results.
>
> ---
>
> [1] D. Mellinger and V. Kumar, “Minimum snap trajectory generation and control for quadrotors,” in 2011 IEEE Int. Conf. on Robot. and Automat., 2011, pp. 2520–2525.